# Effects of the Inclusion of Fermented Mulberry Leaves and Branches in the Gestational Diet on the Performance and Gut Microbiota of Sows and Their Offspring

**DOI:** 10.3390/microorganisms9030604

**Published:** 2021-03-15

**Authors:** Yuping Zhang, Chang Yin, Martine Schroyen, Nadia Everaert, Teng Ma, Hongfu Zhang

**Affiliations:** 1State Key Laboratory of Animal Nutrition, Institute of Animal Sciences, Chinese Academy of Agricultural Sciences, Beijing 100193, China; zhangyuping@caas.cn (Y.Z.); 82101186178@caas.cn (C.Y.); zhanghongfu@caas.cn (H.Z.); 2Precision Livestock and Nutrition Laboratory, TERRA Teaching and Research Centre, Gembloux Agro-Bio Tech, University of Liège, 5030 Gembloux, Belgium; martine.schroyen@uliege.be (M.S.); nadia.everaert@uliege.be (N.E.)

**Keywords:** mulberry, sows, gestation, lactation, fecal microbiota

## Abstract

Fermented feed mulberry (FFM), being rich in dietary fiber, has not been fully evaluated to be used in sow’s diet. In this study, we investigated the effects of 25.5% FFM supplemented in gestation diets on the performance and gut microbiota of sows and their offspring. Results showed that the serum concentration of glucose, progesterone, and estradiol were not affected by the dietary treatment, while the level of serum insulin and fecal short chain fatty acid were both reduced in FFM group on gestation day 60 (G60, *p* < 0.05). Additionally, FFM increased both voluntary feed intake and weaning litter weight (*p* < 0.05), while decreased the losses of both Backfat thickness and bodyweight throughout lactation (*p* < 0.05). 16S rRNA sequencing showed FFM supplementation significantly increased the diversity and relative abundance of sows’ fecal microbiota on G60 (*p* < 0.05). The differential microbiota for sows from FFM group was that Bacteroidetes was increased on G60 while Firmicutes were decreased on Lactation day 7 (L7, *p* < 0.05), and which for the FFM piglets was that both *unclassified_f_Lachnospiraceae* on L0 and *norank_f_Ruminococcaceae* on L7 were increased (*p* < 0.05). In short, FFM can be recognized as a potential feed ingredient used in sow’s diet.

## 1. Introduction

Dietary fiber (DF) plays an indispensable role in the nutrition of sows due to its potential value in improving performance and intestinal health. Studies have shown that the addition of fiber to the gestational diets of sows can improve satiety [1], decrease stereotypical behaviors [2], reduce the occurrence of prolonged constipation [3], increase the survival rate for embryos in the early gestation stage [4,5], and improve litter performance, including the litter size and weight at birth and weaning [6,7]. Additionally, a high consumption of dietary fiber during gestation might increase food intake during lactation [8,9]. However, studies have reported no or even adverse effects of high-fiber diets on sows’ litter performance and feed intake during lactation, which could be due to the complexity of the fiber sources, structures, and compositions, as well as the proportions, periods, and durations with which the fiber is added to the diet.

The effects of high fiber intakes on the composition of the intestinal microbiota and their metabolites, such as short-chain fatty acids (SCFA), have also received increasing attention [10]. Studies have shown that high-fiber diets can cause changes to the pig intestinal bacteria and positively impact intestinal immune function [11]. SCFAs can directly activate G-protein-coupled receptors, inhibit histone deacetylases, and serve as energy substrates [12]. Additionally, various physiological processes may influence health and disease [13]. In addition to dietary fiber’s direct effects, a new strategy entails adding fiber to the mother’s diet in an attempt to improve the offspring’s microbiota. Recent studies have found that the transmission of microbiota between sows and piglets plays a vital role in establishing the offspring’s microbiota, which begins during the gestation period [14,15]. Previous research found that supplementation with wheat bran during the sow gestational and lactation periods adjusts the microbiota composition of both the sows and piglets, as well as the microbiota in the umbilical cord blood [16].

Mulberries (*Morus alba*), whose leaves are rich in nutrients such as proteins, vitamins, and minerals, are widely distributed throughout China. They are a low-cost plant material often used as a potential feedstuff for livestock, improving animal production performance, feed conversion efficiency, and meat quality [17,18]. They are also rich in active substances, such as 1-deoxynojirimycin, which reduces cholesterol and regulates blood glucose levels [19]. This could help to prevent sows becoming overweight during pregnancy [20]. Studies have shown that supplementation with mulberry leaf polysaccharide (MLP) (92.22% pure and consisting of glucose, mannose, arabinose, galactose, xylose, rhamnose, and ribose in a ratio of 250:66:6:3.25:2.5:1.25:1) could reduce diarrhea in weaned piglets [21].

We hypothesized that supplementation with a high concentration (25.5%) of FFM in a sow’s gestational diet could improve the performance of both the sow and its piglets by adjusting blood hormones related to reproduction and glucose metabolism, colostrum components, and the gut microbiota.

## 2. Materials and Methods

### 2.1. Ethical Approval

This animal experiment protocol was approved by the Ethical Committee of the Chinese Academy of Agriculture Sciences (protocol number: IAS2019-31) and carried out on a commercial pig breeding farm located in Zhumadian, Henan Provence, China.

### 2.2. Fermented Feed Mulberry 

The fermented feed mulberry (FFM) was provided by Henan Shiji Tianyuan Ecological Technology Co., Ltd. (Zhengzhou, Henan, China) The general production process involved chopping, crushing, and mixing the tender leaves and branches of the mulberry feed with 20% corn powder and 10% wheat bran (this component was reduced during the subsequent preparation of fermented mulberry feed) containing 10^6^ (CFU) g^−1^ of lactic acid bacteria. All the samples were stored at 4° Celsius to prevent deterioration. 

### 2.3. Experimental Design, Feeding Regime, and Animals and Housing

Forty multiparous sows (Yorkshire × Landrace; parity, 3–7 (mean ± SEM = 4.4 ± 0.3)) were randomly allocated to two treatment groups, with 20 in each, and one sow and its litter were considered to represent one biological replicate. From gestation day 0 (G0), three days after artificial insemination with the semen from a Duroc boar, until gestation day 110, sows were housed in individual gestation stalls (2.1 × 0.6 m^2^). In the control (CON) group, the sows were fed with a corn-soybean meal diet, while in the FFM group, 25.5% of the corn-soybean meal was replaced with FFM. All the treatment diets were formulated to meet the nutritional requirements of gestating and lactating sows, as recommended by the National Research Council (NRC, Washington, DC, USA, 2012) [22]. Before being used in the gestation diet, the total concentration of viable bacteria in the FFM was 1.3 × 10^5^ per gram. The sows were fed twice per day at 06:30 and 16:30. The two groups of sows received different feed supplies to ensure that they received similar digestible energy (DE) intakes (Table 1). On G110, the sows were transferred into farrowing rooms with individual farrowing crates (2.1 × 1.5 m^2^). During the lactation period, all of the sows were fed the same lactation diet. The quantity of the feed given was 0.5 kg on the parturition day, and this was gradually increased by 1.0 kg/d until the 5th day of lactation, when food was provided ad libitum. The sows had free access to water throughout the experimental period. Within 24 h of farrowing, the litter size was standardized to approximately 10 piglets by cross-fostering within the treatment. The piglets were weaned at 21 d and had no access to creep feed during the suckling period. The timeline, different sampling points, numbers of samples, and parameters for the sows and piglets in both the CON and FFM groups are shown in Figure 1.

### 2.4. The Nutritional Indicators of the FFM and Diet

The FFM and diets was milled through a 1 mm screen (Christy and Norris Hammer Mill, Chelmsford, England). The dry matter content was determined after drying in the Constant-Temperature Drying Oven (BIOBASE Group, Jinan, China) at 105 °C for 5 h, according to Association of Official Agricultural Chemists (AOAC), 2007; method AOAC.930.15. Crude ash content (AOAC.942.05, 2005) was determined after ignition of a weighted sample in a muffle furnace (Nabertherm, Bremen, Germany) at 550 °C for 6 h. The gross energy content was determined using an adiabatic bomb calorimeter (Parr Instruments, Moline, IL, USA). The nitrogen content was determined using the N determination with Kjeltec Analyzer (Hanon Instruments, Jinan, China) (AOAC.976.05, 2007). The AOAC (AOAC.962.09, 2005) method was used for determination of crude fiber. The NDF and ADF were calculated using an ANKOM A2000i (ANKOM Technology Corporation, Macedon NY, USA), with the reagents of Neutral detergent solution concentrate, Acid detergent solution concentrate, and the α-amylase concentrate (ANKOM Technology Corporation, Macedon, NY, USA), according to Van Soest et al. (1991) [23]. The total dietary fiber (TDF) and IDF were determined according to AOAC2007 (methods 985.29 and 991.43, respectively), while the SDF was calculated as the TDF value minus the IDF, with the Total Dietary Fiber Assay, TDF-100A (Sigma-Aldrich, St. Louis, MO, USA). The DE of the FFM for pigs was determined by the regression method (Adeola, 2001) [24]. The DE of the gestation diets for both the CON and FFM groups was calculated as the sum of the DE content of each ingredient, except the FFM, which was calculated as the DE value for pigs of each component multiplied by its proportion in the diet. The DE values for pigs of all the other ingredients were taken from NRC 2012 [22].

### 2.5. Performance Parameters

The backfat thicknesses (BF) of the sows were recorded on G0, G20, G60, and G108 and at weaning. The BF was estimated at 7 cm from the backbone in a straight line from the tip of the last rib (P2) using ultrasound (PIGLOG105, SFAK-Technology, A mode scanner, SFK Technology A/S Helver, Denmark). The BWs of the sows were recorded at G108 and weaning. Within 12 h of farrowing, the number and weight of live piglets born per litter were recorded. The litter size and weight were also recorded at the moment of weaning. During lactation, the sows were fed ad libitum, and the feed intake (FI) was determined at L1, L3, L7, L14, and L21. The daily VFIs of the sows were measured by calculating the daily amounts of feed supplied minus the amounts refused. The weaning-to-estrus interval (WEI), constipation rate in the sows, and diarrhea rate in the newborn piglets were all measured and recorded.

### 2.6. Blood, Feces, and Colostrum Sampling

Fasting blood samples (10 mL) were collected via the ear vein from each sow before the morning meal on G0, G20, and G60. The fresh blood was placed in 5 mL sterile, evacuated blood collection tubes that were then left to stand for 30 min and subsequently centrifuged at 3000 g for 10 min. The serum was stored in a freezer at −20 °C until further analysis. An assay kit (E1010, Applygen Technologies, Beijing) was used to determine the serum glucose concentrations using the glucose oxidase (GOD) method. The insulin concentrations were measured with the Invitrogen Porcine Insulin ELISA Kit (Thermo Fisher Scientific, Waltham, MA, USA). The estradiol profiles were determined with an estradiol (E2) enzyme immunoassay test kit (Oxis International, Foster City, CA, USA), while the progesterone profiles were determined with the KIP1458 kit (Diasource, Ottignies, Belgium).

On G60, the sows’ fresh feces were collected, snap-frozen in liquid nitrogen, and stored in a −80 °C freezer until further analysis. Gas chromatography was used to determine the short-chain fatty acid (SCFA) concentrations in the feces on G60, as described by Chen et al. (2013), with minor modifications, using an Agilent 7820 gas chromatograph with an autosampler. We used two fecal scoring standards to score the feces from the sows (from 60 to 66 days of gestation and the first week of lactation) and piglets (the first week of birth). The sow feces were assessed according to the Bristol stool scale [25], on which 0 indicates an absence of feces, 1 indicates dry feces and pellets, 2 indicates dry to normal, 3 indicates normal and soft, 4 indicates between normal and wet, and 5 indicates very wet feces. Those with fecal scores < 3 were recorded as being constipated, and the constipation rate for each group was recorded. The diarrhea rate for the piglets from lactation days 3 to 10 was determined according to four levels of scoring: 0, normal; 1, pasty; 2, semiliquid; and 3, liquid [26]. Those with fecal scores > 2 were recorded as having diarrhea. As shown in Figure 1, 7 samples were assessed for each treatment group in each period. Hence, feces samples from 28 sows and 28 piglets were used for 16S rRNA sequencing.

Within the first 24 h of farrowing, 15 mL samples of colostrum were manually collected from functional tits to determine the concentrations of proteins, fats, lactose, milk solids, and nonfat milk solids by Fourier transform infrared spectroscopy using an automatic Standard Lactoscope FT-MIR (Delta Instruments, Drachten, The Netherlands), as previously performed by McParland et al. (2016) [27]. The predictive models provided by the manufacturer were originally designed for the analysis of cow milk and were consequently updated for sow milk by a slope and bias correction using a set of sow milk samples with known reference values obtained by chemical reference analysis. 

### 2.7. Statistical Analyses

The data on the sows’ BFs and BWs; concentrations of glucose, insulin, estradiol, and progesterone in the sows’ sera; contents of proteins, fats, lactose, and milk solids in the sows’ colostrum; the SCFA profiles of the sows’ feces; the litter sizes; and the BWs of the piglets at birth and weaning were all analyzed with SAS 9.4 (Institute, Cary, NC, USA). The data are presented as means ± SEMs, and significant differences were considered at a probability level of *p* < 0.05.

### 2.8. DNA Extraction, 16S rRNA Gene Amplification, and Sequencing and Analysis

Microbial community genomic DNA was extracted with the QIAamp DNA Stool Mini Kit (Qiagen, Düsseldorf, Germany), in accordance with the manufacturers’ protocol. The 341F and 805R primers (5′-CCTACGGGNGGCWGCA-3′ and 5′-GACTACHVGGGTATCTAATCC-3′, respectively) were used to amplify the V3–V4 hypervariable region of the 16S rRNA gene. The purified amplicons were sequenced on an Illumina MiSeq PE300 platform (Illumina, San Diego, USA). The raw reads were deposited into the NCBI Sequence Read Archive (SRA) database (accession number: PRJNA694696).

The sequencing reads were demultiplexed, quality filtered with fastp version 0.20.0, and merged with FLASH version 1.2.7. Operational taxonomic units (OTUs) were clustered with a 97% similarity cutoff using Uparse (version 7.1, http://drive5.com/uparse/, accessed on 24 February 2021) [28], and chimeric sequences were identified and removed. The sequence reads from the different samples varied in number (mean ± SEM of 25,554 ± 7125 for sows’ feces and 20,687 ± 7488 for piglets’ feces) and were normalized to 8088 reads to make them comparable. The taxonomy of each representative OUT sequence was analyzed with RDP Classifier version 2.2 [29] using the Silva 132 reference database with a confidence threshold of 0.7. Mothurb.1.30.1 was used to perform an alpha diversity analysis and unweighted principal coordinate analysis (PCoA) [30]. The DESeq2 software package was used to measure the dietary differences in individual OTUs at the different taxon levels [31,32]. The predictive functional profiling of the microbial communities was conducted by a Phylogenetic Investigation of Communities by Reconstruction of Unobserved States (PIRCUSt) [30]. A Benjamini–Hochberg corrected *p*-value (false discovery rate, FDR) of 0.05 was considered statistically significant [33].

## 3. Results

### 3.1. Sows’ Reproductive Performance

As shown in Figure 2, the voluntary feed intake (VFI) throughout the lactation period was significantly increased by FFM supplementation, mostly on L1, L3, and L21 (*p* < 0.05, Figure 2a). There was no difference in backfat thickness (BF) between the FFM and CON groups on gestation day 0, 20, 60, or 110 (*p* > 0.05, Figure 2b). Throughout the lactation period, the loss of BF and BW the sows in the FFM group were significantly lower than those in the CON group (*p* < 0.05).The mean ± SEM of BF values (centimeter) on L21 for the CON and FFM groups were 20.45 ± 0.68 and 18.85 ± 0.39 (*p* < 0.05). The loss of BW (kilogram) for the FFM sows was significantly less than that for those in the CON group (mean ± SEM of CON vs. FFM: 32.58 ± 2.68 vs. 23.20 ± 2.64; *p* < 0.05, Figure 2f).

The serum and colostrum evaluations revealed that the FFM treatment affected the insulin level and colostrum components (Figure 3). The serum concentrations of glucose (Figure 3a), progesterone (Figure 3b), estradiol (Figure 3c), and insulin (Figure 3d) in the sows were not affected by the dietary treatment during gestation, except for the serum insulin level, which was significantly lower in the FFM sows on gestation day 60 than in the CON sows (*p < 0.01*). Compared to in the CON group, the milk protein (*p* = 0.07), fat (*p* = 0.24), and lactose (*p* = 0.24) concentrations in the colostrum were not affected by the FFM treatment, while the total milk solid concentration (*p* < 0.05, Figure 3e) and concentration of milk solids excluding fat (*p* < 0.05, Figure 3e) in the FFM pigs were significantly lower than in the CON pigs. Nevertheless, the piglets’ bodyweights during weaning increased dramatically in the FFM group (Figure 1f; *p* < 0.05). The WEIs of the FFM sows showed a trend toward being longer than those of the CON sows (mean ± SEM of 7.35 ± 1.14 days for FFM vs. 6.73 ± 1.03 days for CON, Figure 1g; *p* > 0.05, Figure 3f). The constipation rate for the sows during gestation was 21.05 % in the CON group vs. 0% in the FFM group, while during the first week of lactation, the constipation rate for the CON sows was as high as 47.06%. However, it remained 0 % in the FFM group. From days 3 to 10, 13.3% of the newborn CON piglets had diarrhea, but none of the FFM newborns did.

### 3.2. Serum Glucose and Hormone Concentrations, Components in the Colostrum, and Fecal SCFAs

The sows’ serum concentrations of glucose (a), progesterone (b), estradiol (c), and insulin (d) were not affected by dietary treatment on G0, G20, or G60 (*p* > 0.05). However, serum insulin was significantly lower on G60 in the FFM sows than in the CON sows (*p* < 0.01) (Figure 2). On G60, the concentrations of all the tested SCFAs (f), including acetic, propionic, isobutyric, butyric, isovaleric, and valeric acid, were significantly lower in the feces of the FFM sows than in the CON sows (*p* < 0.01). By contrast, the concentrations of total milk solids and total milk solids excluding fat (e) were significantly lower in the FFM group than in the CON group (*p* < 0.05).

### 3.3. Fecal Microbiota

#### 3.3.1. Sows’ Fecal Microbiota Composition

The alpha diversity (Sobs, Chao1, and ACE) and diversity indices (Shannon and Simpson) of the fecal microbiota are shown in Table 2. On G60, compared with the CON group, the Sobs, ACE, and Chao 1 indices were significantly increased and the Simpson’s index was significantly reduced in the FFM group (*p* < 0.05). On L7, there were no differences between the CON and FFM groups for most of the indices measured. The Good’s coverage values for the sows and piglets were all above 97%, suggesting that the present study captured the dominant phylotypes. Unweighted UniFrac metrics were used to measure the beta diversity, and the distances were visualized by a principal coordinate analysis (PCoA, Figure 4a). The results show that the sows’ microbiota communities differed on G60 between the treatment groups; however, there was no clear separation between the groups on L7.

As shown in Table 3, the differences in the compositions of the fecal microbiota of the sows fed the CON and FFM diets were compared on G60 and L7. The dominant phyla in the pigs from both treatment groups were Firmicutes, Bacteroidetes, Spirochaetes, and Proteobacteria, covering more than 90% of each treatment group’s total taxa. The relative abundance of the Bacteroidetes phylum was significantly higher in the FFM group than it was in the CON group on G60 (*p* < 0.05, *q* > 0.05). However, on L7, when each group was fed the same lactation diet without FFM, the relative abundance of the Firmicutes phylum was lower in the feces of the FFM sows than the CON sows (*p* < 0.05, *q* > 0.05). On G60, within Firmicutes, there were five genera that remarkably increased in concentration in the FFM group compared to the CON group: *Ruminococcaceae_NK4A214_group* (*p* < 0.05, *q* > 0.05), *Ruminococcaceae_UCG005* (*p* = 0.01, *q* = 0.07), *Family_XIII_AD3011_group* (*p* = 0.01, *q* = 0.046), *Ruminococcaceae_UCG-013* (*p* = 0.047, *q* = 0.15), and *Lachnospiraceae_AC2044_group* (*p* = 0.01, *q* = 0.07). In addition, within Bacteroidetes, the concentration of the *Prevotellaceae_UCG-001* genera were increased, while that of *Parabacteroides* was reduced, in the FFM group compared to the CON group (*p < 005, q < 0.05*) when considering the p-values. On L7, there were two genera, *Prevotellaceae_NK3B31_group* (*p* = 0.04, *q* > 0.05) and *norank_f_p-2534-18B5_gut_group* (*p* = 0.04, *q* > 0.05), that showed significantly higher concentrations in the FFM group compared to the CON group. By contrast, two genera, *Family_XIII_AD3011_group* (*p* = 0.02, *q* > 0.05) and *Candidatus_Soleaferrea* (*p* = 0.03, *q* > 0.05), were markedly less abundant in the FFM group when compared to the CON group (*p* = 0.034, *q* = 0.441), in terms of statistical significance.

We also found that the relative abundance ratios in the sows’ feces showed remarkably stage-specific phenotypes. For instance, at the phylum level, the abundance (mean ± SEM) of Firmicutes increased significantly from G60 to L7. Meanwhile, that of Bacteroidetes declined from G60 to L7.

#### 3.3.2. Piglets’ Fecal Microbiota

The α diversity (Table 4) for the piglets on L0 and L7 did not show any effects of FFM treatment (*p* > 0.05). The PCoA results (Figure 4b) illustrate the piglets’ bacterial communities, which were obviously separated at L0 and L7. Neither L0 nor L7 were split clearly between groups. In general, the distances of the samples of piglets fed by sows on the CON and FFM diets showed highly similar bacterial compositions. 

As shown in Table 5, the differences in the composition of the fecal microbiota of the piglets birthed by the CON and FFM sows were compared on L0 and L7. The dominant phyla on L0 were Firmicutes, Actinobacteria, and Proteobacteria, which accounted for more than 90% of the total taxa in the pigs from each treatment group in each period. None of them showed differences between the treatment groups. Within Firmicutes, the relevant abundance of fecal genera of *unclassified_f_Lachnospiraceae* was significantly higher in the FFM piglets than in the CON ones (*p* = 0.03, *q* > 0.05). On L7 (Table 6), the dominant phyla were Firmicutes, Proteobacteria, Actinobacteria, and Bacteroidetes, which covered more than 90% of the total taxa in the pigs from each treatment group in each period, and no differences were observable between the treatment groups. Within Firmicutes, the relative abundance of *norank_f_Ruminococcaceae* was significantly higher in the FFM piglets than in the CON ones (*p* = 0.046, *q* > 0.05). Similarly, to for the sows, we also found that the relative abundance ratios in the piglets’ feces were remarkably stage-specific. For instance, at the phylum level, the abundance (mean ± SEM) of Firmicutes (CO*N* = 76.12 ± 3.92% and FFM = 62.82 ± 11.74%) decreased from L0 to L7 (CO*N* = 59.38 ± 5.81% and FFM = 58.83 ± 7.06%), while that of Bacteroidetes (CO*N* = 0.99 ± 0.23% and FFM = 2.71 ± 1.46%) increased from L0 to L7 (CO*N* = 18.64 ± 6.78% and FFM = 11 ± 2.21%). 

#### 3.3.3. Predicted Functional Changes in the Fecal Microbial Community on G60

The results show that only the microbial community functions related to carbohydrate and lipid metabolism were significantly affected by dietary FFM supplementation (Table 7). The top seven pathways concerning carbohydrate metabolism are listed in the table. The FFM group showed a significantly lower abundance of the Kyoto Encyclopedia of Genes and Genomes (KEGG) pathways “Glycolysis/Gluconeogenesis” and “Citrate cycle (TCA cycle) metabolism” (*p < 0.05*) and a remarkably higher abundance of the KEGG pathways “pentose and glucuronate interconversions” and “fructose and mannose metabolism” (*p* < 0.05). Three of the top seven pathways concerning lipid metabolism were increased in the FFM group: “fatty acid biosynthesis, “fatty acid elongation”, and” secondary bile acid biosynthesis” (*p* < 0.05). By contrast, only the “synthesis and degradation of ketone bodies” pathway was reduced (*p* < 0.05). None of the pathways related to the metabolism of amino acids or energy differed significantly between the treatment groups (*p* > 0.05).

## 4. Discussion

Optimizing the reproductive performance of sows will play a vital role in the improvement of the pig industry. In recent years, the administration of high-fiber diets to sows has attracted increased attention, mainly due to the potential benefit for sow performance. The possible benefits of a high-fiber diet include impacts on serum hormone levels, milk components, the intestinal microbiota, and immune function [34,35,36,37,38,39]. This study aimed to explore the effects of fermented mulberry, which has both a high fiber content and 1.3 × 10^5^ lactic acid bacteria per gram, on the performance of sows and investigate the potential mechanisms of action.

### 4.1. Sows’ Performance

Our results show that the supplementation of feed with 25.5% FFM during the gestation period improved the sows’ performance. During the lactation period, voluntary feed intake was increased, bodyweight loss was reduced, and the weaning weights of the piglets were increased on L21. These results are consistent with those from another study by Shang et al., where supplementation with 20% sugar beet pulp (SBP) during gestation and 10% SBP during lactation was examined [40,41]. Weng et al. showed positive effects on performance when 20% soy hulls (SH) were added during pregnancy and lactation [42]. Tan et al. reported that sows fed a konjac flour diet consumed more during lactation and that piglets were significantly heavier at weaning [7]. Mroz et al. observed that increasing the concentration of oat hulls from 0 to 50% in the gestational diet linearly increased the litter weight [43]. However, Leblois et al. found that supplementation with 25% wheat bran (WB) during gestation and 14% during lactation slightly reduced sows’ feed intake during lactation but did not affect the litter weight gain [44]. To summarize the above results, it has been shown that the inclusion of both soluble and insoluble fiber in the gestational diets of sows may increase the feed intake of the sows during lactation and affect the litter performance of the piglets. Sows usually lose some BF and BW during lactation due to maternal effects whereby nutrients are prioritized for lactation purposes while feeding piglets. This study found that sows supplemented with FFM had less BF and BW loss compared with those in the CON group. Additionally, the sows that lost excessive amounts of BW had extended weaning-to-estrous intervals (WEIs) [45]. The tendency for prolonged WEIs in the FFM group in this study suggests that the amount of fiber added may have been close to the maximum a sow can tolerate. 

### 4.2. Short-Chain Fatty Acid Concentration in Feces, and Components in Serum and Colostrum

Previous studies have shown that the consumption of a high-fiber diet could change the SCFA profile in the digesta of the different parts of the intestine [46]. However, there have often been inconsistent conclusions. For example, in a study by Shang et al., it was observed that sows supplemented with sugar beet pulp had increased levels of acetic acid, butyric acid, and total SCFAs [41]. By contrast, sows supplemented with WB only showed an increase in butyric acid [40]. In this study, the concentrations of acetic, propionic, butyric, valeric, and isovaleric acid were all significantly lower in the feces of the FFM sows than those of the CON sows at G60. The SCFA profile in the feces indicates the amounts produced by microbial fermentation minus the amount absorbed by the intestinal wall and the amount consumed by microorganisms. In this study, the mulberry materials used were fermented by *Lactobacillus* and *Saccharomycetes* in vitro; the fermentability of the FFM was relatively low. When added into the sows’ diets at a concentration of 25.5% on a dry matter basis, the feed’s fermentability in the hindgut was lower than that of the feed consumed by the control group. Secondly, the addition of fermented mulberry significantly increased the richness of the sows’ fecal microbiota. SCFAs could be used to generate carbon scaffolds in microbial cells. With an increasing concentration of gut microbiota, the consumption of SCFAs by the microbiota also increased, further exacerbating the decrease in SCFA. Previous in vitro studies have shown that acetate is utilized by *Faecalibacterium prausnitzii* and *Roseburia sp*. [47,48]. In the present study, we found the genus of the *Roseburia* to be significantly greater in the FFM group on G60. 

The serum insulin levels of the sows on G60 were significantly lower in the FFM group than in the CON group. Previous studies have shown that sows fed soluble fiber have greater insulin sensitivity, as described by Xu et al., who supplemented sows with 2.0% guar gum plus pregelatinized waxy maize starch [39]. In addition, Tan et al. noted an elevation of insulin sensitivity when konjac flour, a highly soluble fiber, was added to feed [7]. In our study, the concentrations of milk solids and nonfat milk solids in the colostrum were significantly reduced following FFM supplementation. This is consistent with the results of several studies finding that the consumption of a high-fiber diet increased the lipid concentration in the colostrum [44,49]. The reason the addition of mulberry to feed leads to reductions in milk solids and nonfat milk solids in the colostrum requires further research.

### 4.3. Fecal Microbiota

According to previous studies, the consumption of a high-fiber diet can change the composition of the intestinal microbiota, which are affected by the source and type of fiber added. The amount of fiber supplemented also plays an essential role in these effects. Several pig-based studies have shown that excessive fiber addition reduces the abundance and diversity of the intestinal microbiota. Wang et al. found that the richness of the intestinal microbiota in nursing pigs (mean WB = 24 kg) in a group fed a diet supplemented with 15% alfalfa (19% TDF) was significantly lower than that of a control group (14% TDF) [50]. Similarly, Pu et al. reported that many genera had higher relative abundances in the microbiota of finishing pigs fed a diet containing 19.10% TDF than in those fed a diet containing 24.11% TDF (mean bodyweight = 62.9 kg) [51]. This study found that the FFM sows’ fecal microbiota richness (32.1% TDF) was higher than the richness in the CON sows (24.1%TDF). The main reason our results are different from those of the above two studies is the differences in the ages and weights of the pigs; the previous studies used fattening pigs, while we used sows not less than three parities. The average weight of each sow was greater than 250 kg. Studies have shown that adult sows have more developed gastrointestinal systems than young growing pigs, resulting in a superior capacity to digest fibrous diets [52]. Similarly, we speculate that there is a maximal limit to fiber addition for sows. The maximum amount of fiber appropriate for the sow diet could be determined by decreases in the abundance and diversity of the intestinal microbiota. 

In this study, the relative abundance and diversity of sows’ fecal microbiota significantly increased at G60 following treatment with FFM. However, there tended to be a similarity in abundance between the two groups after the diet was changed to the same lactation feed for just one week, suggesting that intestinal microbiota changes are sensitive to gestational diets. Consistent with the results of Wang et al., we also found that the sows’ fecal microbiota had stage-specific characteristics [50]. In this study, with the transition from gestation to lactation, the sows’ most noticeable shifts were decreases in the proportion of Firmicutes and ratio of Bacteroidetes. The change in the abundance of Firmicutes is contrary to the results obtained by Tan et al. [35]. A possible explanation for this is that in their research, the supplementation was performed using 2.2% konjac flour (highly soluble fiber), while our study included 25.5% fermented mulberry containing a high proportion of insoluble fiber. Furthermore, the effects of the physiological stage on the microbiota seem to be more important than the impact of the diet. Similarly, the microbiota in the feces of the piglets were also found to differ with age. The diversity and abundance of the piglets’ feces were significantly greater at L0 than L7. A possible reason for this is that the meconium is affected by the sows’ amniotic fluid and the vaginal, skin, and fecal microbiota. However, Leblois et al., using high proportions of wheat bran (24% in gestation and 15% in lactation), showed differences in the microbial compositions of sows’ feces between the two dietary treatments during gestation, while the differences were less pronounced during lactation. Moreover, no major differences in the compositional structures of the sows’ microbiota were observed between G98 and L20 [16]. The differences between the results of our two studies may be due to the different ages of the pigs studied during lactation. In the present study, even without mulberry during the lactation period, diet still had a particular impact on the sows’ microbiota at L7. Regarding Leblois’ study, even when wheat bran was added to the diet during lactation, there was no effect on sows’ fecal microbiota at L20. This might be because the disturbance of the microbiota by a high-fiber diet is dose-dependent, whereby supplementation with 25% wheat bran in the gestational diet was sufficient but supplementation with 14% wheat bran in the lactation diet was not enough to alter the microbiota composition compared to that of the pigs fed the control diet. Similarly, this study mainly compared G60 with L7 for sows and L0 and L7 for piglets, indicating that the perinatal period may be a critical period for sows and piglets in terms of changing the gut microbiota.

The dominant phyla in the sows’ feces in both treatment groups, regardless of the period, were Firmicutes and Bacteroidetes, which is consistent with other studies [53,54,55,56]. The sows fed FFM showed an increased abundance of the Fibrobacteres, Bacteroidetes, Deferribacteres, and Kiritimatiellaeota phyla and the *Prevotellaceae_UCG-001*, *Family_XIII_AD3011_group*, *Phascolarctobacterium*, and *Ruminococcaceae_UCG-005* genera*. Firmicutes* species are known for their ability to degrade plant polymers in anoxic habitats [57,58,59], and *Bacteroidetes* species are regarded as specialists in the degradation of high-molecular-weight organic matter (i.e., proteins and carbohydrates) [60]. *Prevotella* has been recognized as the primary contributor to the metabolic network involving carbohydrate utilization and the production of SCFA [48] and has been negatively associated with *Escherichia coli*-induced enteric infection in the human colon [61]. Wang et al. and Dou et al. also reported that an increased abundance of *Prevotellaceae_NK3B31_group* helps to alleviate the occurrence of diarrhea in weaned pigs challenged with enterotoxigenic *Escherichia coli K88* [62,63]. Ruminococcaceae is known to produce SCFAs by degrading various polysaccharides and fibers. *Ruminococcaceae_UCG-005* is a genus belonging to the family *Ruminococcaceae* that can ferment indigestible carbohydrates into butyrate [64], and its abundance has been positively correlated with diarrhea incidence and intestinal villus damage [65,66]. As shown in Table 3, the primary identified microbial functional differences were seen in carbohydrate and lipid metabolism at level 3 of the Kyoto Encyclopedia of Genes and Genomes (KEGG) pathways. Overall, supplementation with FFM altered the metabolism of non-starch polysaccharides and synthesis of fatty acids by the microbiota and seemed to promote the cultivation of beneficial bacteria.

During the lactation period, on L7, the sows’ diets were the same. However, the inclusion of FFM still impacted the sows’ microbiota composition, and differences in two genera were noteworthy. First, the abundance of the *prevotellaceae_NK3B31_group* changed from G60 (CON: 3.98 ± 0.54 vs. FFM: 4.09 ± 0.42) to L7 (CON: 0.79 ± 0.14 vs. FFM: 1.33 ± 0.18), and second, the abundance of *Family_XIII_AD3011_group* changed from G60 (Appendix A, CON: 0.81 ± 0.05 vs. FFM: 1.68 ± 0.26) to L7 (Appendix A, CON: 2.19 ± 0.23 vs. FFM: 1.38 ± 0.18). Although the proportion of the genus *prevotellaceae_NK3B31_group* decreased in both treatment groups one week after farrowing, it was always higher in the FFM group than in the CON group. The *prevotellaceae_NK3B31_group* genus has previously been reported to have positive associations with acetic acid, propionic acid, and the total SCFA concentration in the colonic digesta [67]. The presence of the *prevotellaceae-NK3B31* group in the colon may indicate changes in metabolism related to amino acids, carbohydrates, and lipids [68]. The proportion of the *Family_XIII_AD3011_group* in the feces of the CON sows was higher than that in the FFM sows’ feces. *Family_XIII* is commonly found in the gastrointestinal tracts of animals, many strains have been implicated in the production of butyrate, which exerts essential pleiotropic functions [69]. Although studies have shown that the maternal diet’s fiber content could have some effects on the offspring’s microbiota [70], in this study, there were only differences in the abundances of two genera: *unclassified_f_Lachnospiraceae* from L0 (CON: 1.15 ± 0.21 vs. FFM: 0.54 ± 0.15) to L7 (CON: 0.82 ± 0.37 vs. FFM: 0.44 ± 0.21) and *norank_f_Ruminococcaceae* from L0 (CON: 0.53 ± 0.10 vs. FFM: 0.09 ± 0.05) to L7 (CON: 0.22 ± 0.10 vs. FFM: 1.55 ± 0.59). The abundances of *unclassified_f_Lachnospiraceae* and *norank_f_Ruminococcaceae* were associated with the concentration of carbohydrate-active microorganisms [71]. In this study, the concentration of *unclassified_f_Lachnospiraceae* was higher in the meconium (on L0) of the piglets in the CON group. At L7, *norank_f_Ruminococcaceae* was enriched in the feces of the piglets in the FFM group, suggesting that the newborn suckling piglets were affected by the maternal gut microbiota during gestation in the sows that consumed a high FFM diet. Since both sows and piglets are stimulated by their physiology and the environment in the perinatal period, the mechanism by which a high-fiber diet during gestation influences the gut microbiota of sows and their offspring during the subsequent lactation period needs to be studied further.

## 5. Conclusions

Our data suggest FFM as a low-cost local fiber resource for feed that shapes the gut microbial communities of sows. The FFM addition to the gestational diets is a good way to improve sows’ performance in feeding restriction during pregnancy. The most significant benefits were increasing the feed intake during lactation and reducing the rate of constipation during pregnancy and lactation of sows, as well as reducing the diarrhea rate of piglets in the farrowing room, all of above have a positive impact on improving the production performance of sows. The mechanism for these beneficial effects of FFM supplementation may be achieved mainly by adjusting the gut microbiota of the sow and its offspring, rather than through the adjustment of serum reproduction-related and glucose metabolism-related hormones. Besides, the alteration of microbiota caused by gestation FFM supplementation can persist into earlier lactation, providing a more diverse microbial environment to improve intestinal health and mitigate the impacts of harmful microbial. In addition, piglets benefitted with lower diarrhea rate and higher growth in lactation by transferring maternal microbiota. The long-term beneficial effects of fiber diets on sows and its offspring’s gut microbiota and the interaction mechanism of microbiota and sows’ performance should be addressed in future studies.

## Figures and Tables

**Figure 1 microorganisms-09-00604-f001:**
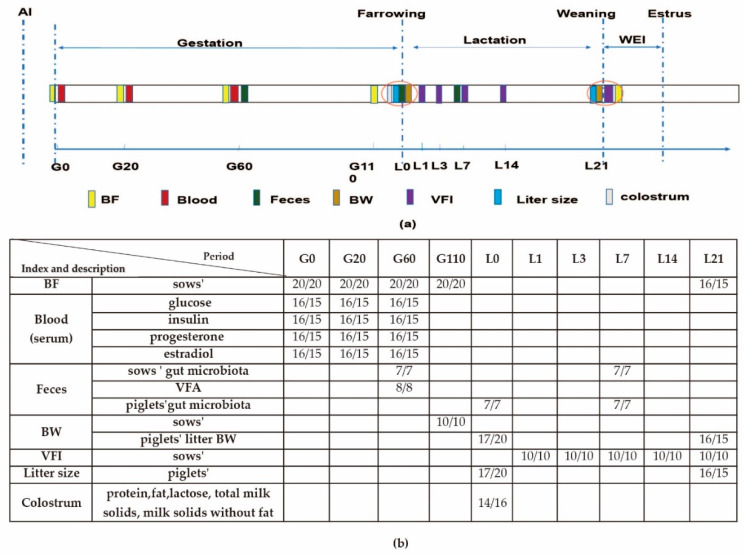
Timeline, numbers of different samples, and indices for sows and their piglets from both treatment groups during the gestation and lactation periods. G0, G20, G60, and G110 indicate gestation days 0, 20, 60, and 80, respectively. Similarly, L0, L1, L3, L7, L14, and L21 represent lactation days 0, 1, 3, 7, 14, and 21, respectively. BF = backfat; BW = bodyweight; VFI = voluntary feed intake. The numbers of each sample are shown as n_CON_/n_FFM_.

**Figure 2 microorganisms-09-00604-f002:**
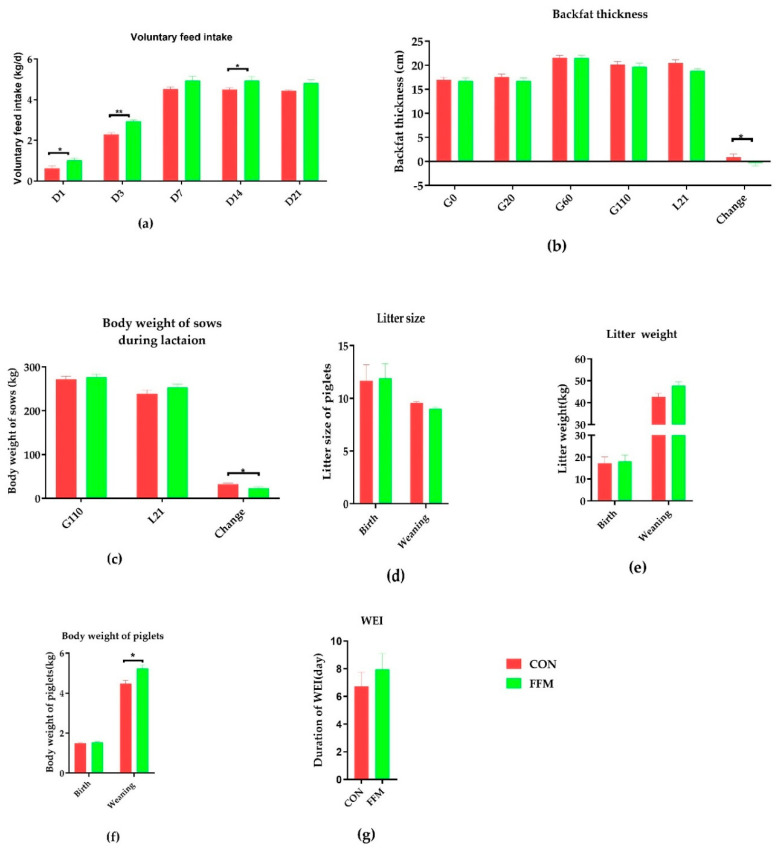
Effects of FFM supplementation on the reproductive performance of sows. (**a**) Daily voluntary feed intake during lactation, *n* = 10. (**b**) Backfat thicknesses on G0, G20, G60, G110, and L21 (*n* = 20). (**c**) Bodyweights on G110 and L21 (*n* = 10). (**d**) Litter sizes at birth and on Day 21 of weaning. (**e**) Litter weights at birth and on Day 21 of weaning. (**f**) Average piglet weights at birth and on Day 21 of weaning. The numbers of sows were as follows: CON (*n* = 17) and FFM (*n* = 20) at birth and CON (*n* = 16) and FFM (*n* = 15) at weaning. (**g**) Weaning and estrus intervals for CON (*n* = 33) and FFM (*n* = 34). CON represents the control diet. FFM represents the diet with 25.5% fermented feed mulberry. *represents *p* < 0.05, and ** represents *p* < 0.01.

**Figure 3 microorganisms-09-00604-f003:**
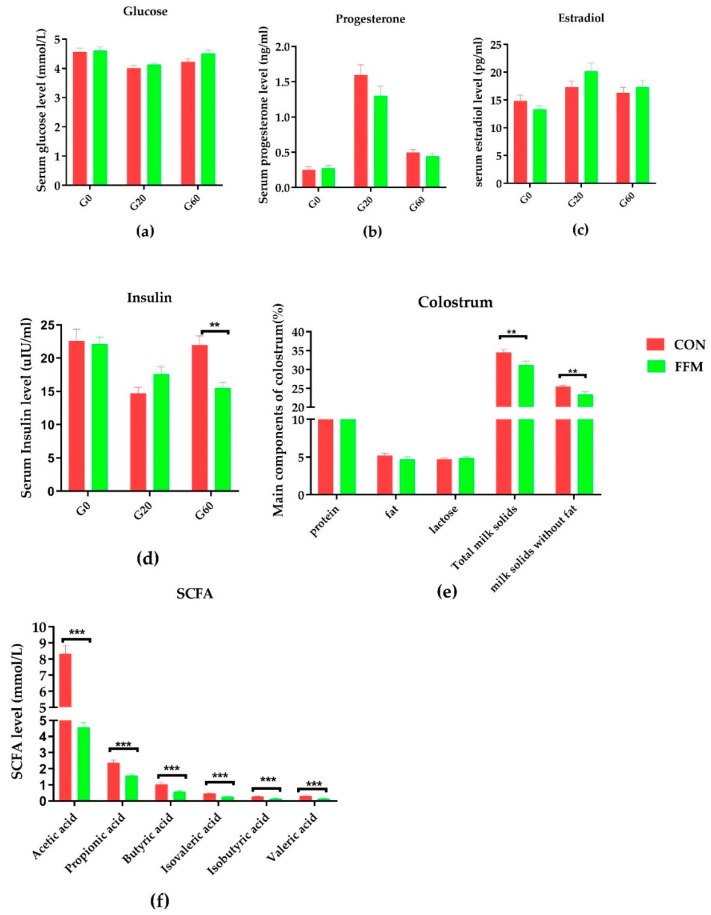
Effects of FFM supplementation on glucose, hormones, and fecal SCFAs. The concentrations of glucose (**a**), progesterone (**b**), insulin (**c**), and estradiol (**d**) in the serum in CON (*n* = 16) and FFM (*n* = 15) sows. G0, G20, and G60 represent gestation days 0, 20, and 60. Components in the colostrum (**e**) for CON (*n* = 14) and FFM (*n* = 16). The short-chain fatty acid (SCFA) concentrations in the feces (*n* = 8) (**f**). CON represents the control diet. FFM represent the diet with 25.5% fermented feed mulberry. TMS means the total milk solid concentration; * represents *p* < 0.05, and ** represents *p* < 0.01, *** represents p < 0.001.

**Figure 4 microorganisms-09-00604-f004:**
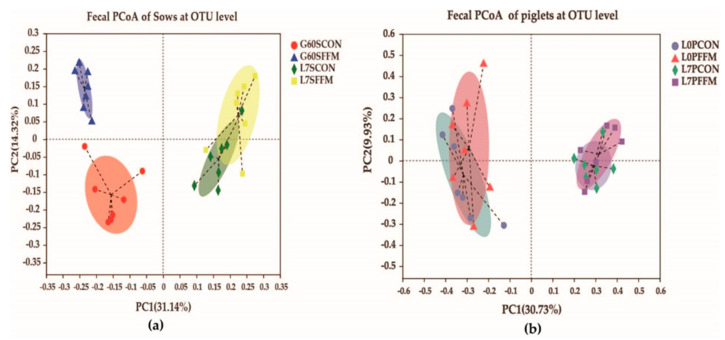
Effects of FFM supplementation shown by the unweighted Unifrac values of the fecal microbiota of sows on G60 and L7 (**a**) and the fecal microbiota of piglets on L0 and L7 (**b**) at the OTU level, as determined by the principal coordinate analysis (PCoA). The Bray-Curtis method was used for the PCoA; different colors or shapes represent different groups’ samples. The closer two sample points are, the more similar the species compositions of the two samples are. N = 7 for each stage of each group. G60SCON = CON sows on G60; G60SFFM = FFM sows on G60; L7SCON = CON sows on L7; L7SFFM = FFM sows on L7; L0PCON = CON piglets on L0; L0PFFM = FFM piglets on L0; L7PCON = CON piglets on L7; L7PFFM = FFM piglets on L7.

**Table 1 microorganisms-09-00604-t001:** Compositions and nutritive value of sow diets and FFM-containing feed and the gestational feeding strategy.

Ingredients (g/kg)	GCON ^3^	GFFM ^3^	LAC ^3^	FFM
Corn	515.4	378.7	515.4	
Soybean meal, 43%	190	140	190	
Fermented soybean meal			80	
Soybean hulls	210	162.5	50	
Wheat flour	25	19.3	25	
Soybean oil	20	15.5	20	
Imported fish meal			30	
Stone powder	9	6	9	
Calcium hydrogen phosphate	12.6	7.9	12.6	
Salt	4.5	2.8	4.5	
Sodium bicarbonate	2	1.3	2	
Choline chloride, 50%	1.4	0.9	1.4	
Vitamin E	0.1	0.1	0.1	
FFM	0	255	0	
Gestation premix ^1^	10	10		
Lactation premix ^2^			60	
Total	1000	1000	1000	
Nutrition and dietary fiber levels ^3^:				
Index	GCON	GFFM	LAC	FFM
Gross energy (MJ/kg)	16.4	16.8	16.6	17.4
Digestible energy (MJ/kg)	13.2	10.3	14.1	9.0
Crude protein %	13.6	14.5	18.9	7.1
Crude fiber%	5.9	14.7	4.4	37.3
Neutral detergent fiber %	23.6	35.3	20.5	52.1
Acid detergent fiber %	8.5	19.5	6.5	42.4
Total dietary fiber %	24.1	32.7	22.3	59.9
Insoluble dietary fiber (IDF, %)	19.1	29.3	16.8	51.5
Soluble dietary fiber (SDF, %)	5	3.4	5.5	8.4
SDF/IDF	3.82	8.62	3.05	6.13
Gestational feeding strategy	Daily feed allowance (kg/d)		Daily DE intake (MJ/d)	
Gestation period	CON	FFM	CON	FFM
G0 to G30	2.3	3	30.36	30.9
G31 to G84	2	2.6	26.4	26.78
G85 to farrowing	3	3.9	39.6	40.17

^1^ Per kilogram of gestation diet provided: Vitamin A, 4000 IU; Vitamin D3, 800 IU; Vitamin E, 40 mg; Vitamin K (menadione), 0.5 mg; biotin, 0.2 mg; folacin, 1.30 g; niacin, 12 mg; pantothenic acid, 12 mg; thiamine, 1.0 mg; Vitamin B6, 1.0 mg; Vitamin B12, 0.015 mg; choline, 1.25 g; iron, 100 ppm; copper, 10 ppm; zinc, 100 ppm; manganese, 25 ppm; iodine, 0.15 ppm; and selenium, 0.15 ppm. ^2^ Per kilogram of lactation diet provided: Vitamin A, 2000 IU; Vitamin D3, 800 IU; Vitamin E, 44 mg; Vitamin K, (menadione) 0.5 mg; biotin, 0.2 mg; folacin, 1.30 g; niacin, 12 mg; pantothenic acid, 12 mg; thiamine, 1.0 mg; Vitamin B6, 1.0 mg; Vitamin B12, 0.015 mg; choline, 1.25 g; iron, 100 ppm; copper, 20 ppm; zinc, 100 ppm; manganese, 25 ppm; iodine, 0.15 ppm; and selenium 0.15 ppm. ^3^ GCON = gestation control diet; GFFM, gestation CON with 25.5% fermented feed mulberry (FFM) diet; LAC, lactation, the same lactation diet.

**Table 2 microorganisms-09-00604-t002:** The α diversity of sows’ feces on G60 and L7.

α Diversity	G60 CON	G60 FFM	*p*	L7 CON	L7 FFM	*p*
Sobs	676 ± 35.26 ^b^	854 ± 22.07 ^a^	<0.001	459.29 ± 26.64	447.71 ± 20.12	NS
Shannon	4.75 ± 0.16 ^b^	5.09 ± 0.06 ^a^	<0.001	4.56 ± 0.09	4.45 ± 0.11	0.056
Simpson	0.02 ± 0.01 ^a^	0.01 ± 0.00 ^b^	0.032	0.02 ± 0.00 ^b^	0.03 ± 0.00 ^a^	0.007
Ace	833.68 ± 32.20 ^b^	996.59 ± 10.83 ^a^	<0.001	681.43 ± 94.10	690.37 ± 111.06	NS
Chao	836.95 ± 27.31 ^b^	993.2 ± 17.88 ^a^	<0.001	632.27 ± 44.43	624.38 ± 41.20	NS
Coverage	0.99 ± 0.00	0.99 ± 0.00	NS	0.98 ± 0.00	0.98 ± 0.00	NS

^a, b^ Means within rows with different superscript letters were significantly different (*p* < 0.05). *n* = 7 for each group. G60SCO*N* = CON sows on G60; G60SFFM = FFM sows on G60; L7SCO*N* = CON sows on L7; L7SFFM = FFM sows on L7; NS = No Statistic.

**Table 3 microorganisms-09-00604-t003:** Compositions of the fecal microbiota of sows on G60 and L7 at the phylum and genus levels.

Phylum/Genus	G60CON	G60FFM	*p*	FDR	L7CON	L7FFM	*p*	*q*
**Firmicutes**	51.89 ± 3.01	53.26 ± 1.92	NS	NS	79.89 ± 2.00	72.93 ± 2.43	0.047	NS
*Christensenellaceae_R-7_group*	13.53 ± 1.88	9.51 ± 1.6	NS	NS	23.2 ± 2.45	21.86 ± 2.44	NS	NS
*Ruminococcaceae_UCG-002*	8.34 ± 0.57	7.4 ± 0.61	NS	NS	9.31 ± 1.56	8.72 ± 0.92	NS	NS
*unclassified_f_Lachnospiraceae*	5.08 ± 0.41	4.46 ± 0.34	NS	NS	8.17 ± 1.48	5.14 ± 0.95	NS	NS
*Lactobacillus*	2.14 ± 0.29	3.24 ± 0.78	NS	NS	4.15 ± 1.08	4.32 ± 0.97	NS	NS
*Ruminococcaceae_NK4A214_group*	1.95 ± 0.27 ^b^	2.71 ± 0.15 ^a^	0.03	NS	3.66 ± 0.35	4.41 ± 0.44	NS	NS
*Ruminococcaceae_UCG-005*	2.41 ± 0.43 ^b^	4.13 ± 0.4 ^a^	0.01	0.07	3.92 ± 0.41	4.9 ± 0.42	NS	NS
*Ruminococcaceae_UCG-014*	2.65 ± 0.5	1.91 ± 0.14	NS	NS	3.81 ± 0.59	3.3 ± 0.48	NS	NS
*unclassified_f_Ruminococcaceae*	1.15 ± 0.14	1.53 ± 0.18	NS	NS	1.77 ± 0.04	1.85 ± 0.22	NS	NS
*Ruminococcaceae_UCG-010*	0.94 ± 0.12	0.82 ± 0.05	NS	NS	0.89 ± 0.11	0.99 ± 0.08	NS	NS
*Family_XIII_AD3011_group*	0.81 ± 0.05 ^b^	1.68 ± 0.26 ^a^	0.01	0.046	2.19 ± 0.23	1.38 ± 0.18	0.02	NS
*Candidatus_Soleaferrea*	0.6 ± 0.17	0.5 ± 0.05	NS	NS	1.03 ± 0.16 ^a^	0.53 ± 0.13	0.03	NS
*Ruminococcaceae_UCG-013*	0.48 ± 0.06 ^b^	0.63 ± 0.04 ^a^	0.047	0.15	0.38 ± 0.05	0.46 ± 0.11	NS	NS
*Clostridium_sensu_stricto_1*	0.44 ± 0.1	0.44 ± 0.09	NS	NS	0.88 ± 0.11	0.59 ± 0.13	NS	NS
*Lachnospiraceae_AC2044_group*	0.59 ± 0.09	0.98 ± 0.1	0.01	0.07	0.67 ± 0.08	0.41 ± 0.1	0.07	NS
*Ruminiclostridium_6*	0.46 ± 0.1	0.29 ± 0.04	NS	NS	0.47 ± 0.11	0.49 ± 0.06	NS	NS
**Bacteroidetes**	27.8 ± 2.35	34.6 ± 1.34	0.03	0.13	7.02 ± 0.78	6.95 ± 0.86	NS	NS
*Rikenellaceae_RC9_gut_group*	6.71 ± 0.81	7.18 ± 0.44	NS	NS	1.62 ± 0.43	0.77 ± 0.07	0.07	NS
*Prevotellaceae_NK3B31_group*	3.98 ± 0.54	4.09 ± 0.42	NS	NS	0.79 ± 0.14	1.33 ± 0.18	0.04	NS
*norank_f_p-2534-18B5_gut_group*	1.6 ± 0.58	2.89 ± 0.69	NS	NS	0.16 ± 0.04	0.36 ± 0.08	0.04	NS
*norank_f_Muribaculaceae*	3.24 ± 0.66	4.31 ± 0.55	NS	NS	1.99 ± 0.69	2.44 ± 0.43	NS	NS
*Prevotellaceae_UCG-001*	2.45 ± 0.56 ^b^	6.69 ± 0.32 ^a^	<0.001	<0.001	0.37 ± 0.16	0.10 ± 0.05	NS	NS
*Prevotella_1*	1.44 ± 0.28	1.9 ± 0.25	NS	NS	0.2 ± 0.04	0.31 ± 0.06	NS	NS
*Parabacteroides*	2.2 ± 0.39 ^a^	0.16 ± 0.02 ^b^	<0.001	0.01	0.37 ± 0.09	0.14 ± 0.03	0.03	NS
*Bacteroides*	2.61 ± 0.59	1.36 ± 0.43	NS	NS	0.2 ± 0.06	0.08 ± 0.01	0.07	NS
**Spirochaetes**	10.67 ± 1.93	6.31 ± 0.72	NS	NS	3.19 ± 0.75	3.66 ± 0.94	NS	NS
*Treponema_2*	10.49 ± 1.96	6.11 ± 0.69	NS	NS	3.13 ± 0.75	3.6 ± 0.94	NS	NS
**Proteobacteria**	2.87 ± 0.49	1.82 ± 0.39	NS	NS	2.58 ± 0.76	5.85 ± 1.39	0.06	NS
*Escherichia-Shigella*	1.88 ± 0.48 ^a^	0.83 ± 0.35 ^b^	0.06	NS	2.46 ± 0.77	5.71 ± 1.35	0.06	NS

^a, b^ Means within rows with different superscript letters are significantly different (*p < 0.05* or *q < 0.05*). *n* = 7 for each group. G60SCO*N* = CON sows on G60; G60SFFM = FFM sows on G60; L7SCO*N* = CON sows on G7; L7SFFM = FFM sows on L7; NS= No Statistic.

**Table 4 microorganisms-09-00604-t004:** α diversity values of piglets’ feces on L0 and L7.

α Diversity	L0PCON	L0PFFM	*p*	L7PCON	L7PFFM	*p*
Sobs	399.43 ± 70.52	356.43 ± 133.92	NS	103 ± 23.19	82.57 ± 22.23	NS
Shannon	3.99 ± 0.13	3.63 ± 0.52	NS	2.84 ± 0.44	2.52 ± 0.48	NS
Simpson	0.05 ± 0.01	0.08 ± 0.04	NS	0.12 ± 0.06	0.17 ± 0.08	NS
Ace	607.6 ± 155.9	481.1 ± 199.92	NS	134.34 ± 25.27	111.17 ± 46.58	NS
Chao	559.1 ± 134.87	473.23 ± 192.29	NS	127.28 ± 27.51	110.31 ± 39.76	NS
Coverage	0.98 ± 0.00	0.99 ± 0.01	NS	1.00 ± 0.00	1.00 ± 0.00	NS

L0PCO*N* = CON piglets on L0; L0PFFM = FFM piglets on L0; L7PCO*N* = CON piglets on L7; L7PFFM = FFM piglets on L7; *n* = 7 for each group. NS = No Statistic.

**Table 5 microorganisms-09-00604-t005:** Composition of the fecal microbiota of piglets on L0 at the phylum and genus levels.

Phylum/Genus	L0PFFM	L0PCON	*p*	*q*
**Firmicutes**	76.12 ± 3.92	62.82 ± 11.74	NS	NS
*Lactobacillus*	14.15 ± 4.759	19.38 ± 4.044	NS	NS
*Streptococcus*	5.191 ± 2.501	7.312 ± 2.149	NS	NS
*Terrisporobacter*	4.983 ± 1.245	5.567 ± 1.533	NS	NS
*Clostridium_sensu_stricto_1*	6.221 ± 1.752	3.976 ± 0.931	NS	NS
*Christensenellaceae_R-7_group*	1.058 ± 0.553	5.958 ± 4.241	NS	NS
*Anaerococcus*	4.269 ± 3.005	2.503 ± 1.973	NS	NS
*Aerococcus*	2.24 ± 0.56	2.381 ± 0.529	NS	NS
*Family_XIII_AD3011_group*	1.457 ± 0.951	2.088 ± 1.119	NS	NS
*Jeotgalicoccus*	2.176 ± 1.423	1.263 ± 0.479	NS	NS
*Facklamia*	0.834 ± 0.487	1.777 ± 0.796	NS	NS
*Gallicola*	2.125 ± 2.092	0.45 ± 0.387	NS	NS
*Peptoniphilus*	1.493 ± 1.382	0.901 ± 0.722	NS	NS
*Ruminococcaceae_UCG-005*	1.547 ± 0.68	0.664 ± 0.151	NS	NS
*norank_f_Lachnospiraceae*	0.726 ± 0.216	1.425 ± 0.486	NS	NS
*Ignavigranum*	0.523 ± 0.396	1.616 ± 0.651	NS	NS
*Turicibacter*	1.212 ± 0.355	0.774 ± 0.284	NS	NS
*unclassified_f_Lachnospiraceae*	0.537 ± 0.149 ^b^	1.15 ± 0.206 ^a^	0.03	NS
*Staphylococcus*	0.475 ± 0.062	1.171 ± 0.428	NS	NS
*Ruminococcaceae_UCG-002*	0.491 ± 0.206	1.077 ± 0.331	NS	NS
*unclassified_f_Ruminococcaceae*	0.489 ± 0.242	0.968 ± 0.387	NS	NS
*Lachnospiraceae_XPB1014_group*	0.606 ± 0.311	0.836 ± 0.203	NS	NS
***Actinobacteria***	8.98 ± 2.55	7.18 ± 3.20	NS	NS
*Rhodococcus*	3.676 ± 2.812	3.469 ± 2.093	NS	NS
*Brevibacterium*	0.682 ± 0.335	1.009 ± 0.558	NS	NS
*Rothia*	0.933 ± 0.419	0.703 ± 0.342	NS	NS
***Proteobacteria***	8.74 ± 2.99	22.76 ± 10.71	NS	NS
*Acinetobacter*	7.579 ± 4.328	1.243 ± 0.522	NS	NS
*Delftia*	5.097 ± 3.199	2.146 ± 0.733	NS	NS
*Brevundimonas*	5.037 ± 3.644	1.526 ± 0.607	NS	NS
*Actinobacillus*	0.214 ± 0.092	1.353 ± 1.213	NS	NS

^a, b^ Means within rows with different superscript letters indicate significant differences (*p* < 0.05 or *q* < 0.05). *n* = 7 for each group. L7PFFM = FFM piglets on L7; NS = No Statistic.

**Table 6 microorganisms-09-00604-t006:** Composition of the fecal microbiota of piglets on L7 at the phylum and genus levels.

Phylum/Genus	L7PCON	L7PFFM	*p*	*q*
**Firmicutes**	59.38 ± 5.81	58.83 ± 7.06	NS	NS
*Lactobacillus*	20.14 ± 4.48	27.34 ± 6.19	NS	NS
*Clostridium_sensu_stricto_1*	4.99 ± 1.52	7.99 ± 2.84	NS	NS
*Streptococcus*	6.45 ± 4.7	1.55 ± 0.42	NS	NS
*Clostridium_sensu_stricto_2*	2.29 ± 0.66	2.62 ± 0.66	NS	NS
*Ruminococcus_2*	2.09 ± 0.63	2.61 ± 0.91	NS	NS
*[Ruminococcus]_gnavus_group*	1.89 ± 0.84	2.71 ± 0.81	NS	NS
*Tyzzerella*	2.22 ± 1.03	1.35 ± 1.23	NS	NS
*[Eubacterium]_coprostanoligenes_group*	2.34 ± 1.09	1.13 ± 0.51	NS	NS
*Ruminococcaceae_NK4A214_group*	2.43 ± 1.88	0.46 ± 0.24	NS	NS
*Lachnoclostridium*	1.24 ± 0.28	0.92 ± 0.42	NS	NS
*Parabacteroides*	1.73 ± 1.28	0.42 ± 0.18	NS	NS
*[Eubacterium]_fissicatena_group*	1.65 ± 0.98	0.36 ± 0.15	NS	NS
*Ruminococcaceae_UCG-004*	1.01 ± 0.47	0.92 ± 0.22	NS	NS
*[Ruminococcus]_torques_group*	0.19 ± 0.09	1.64 ± 1.09	NS	NS
*norank_f_Ruminococcaceae*	0.22 ± 0.1	1.55 ± 0.59	0.046	NS
*Veillonella*	0.84 ± 0.63	0.49 ± 0.39	NS	NS
*unclassified_f_Lachnospiraceae*	0.82 ± 0.37	0.44 ± 0.21	NS	NS
*Peptostreptococcus*	0.59 ± 0.37	0.52 ± 0.26	NS	NS
*norank_f_Lachnospiraceae*	0.61 ± 0.21	0.48 ± 0.18	NS	NS
*Christensenellaceae_R-7_group*	0.75 ± 0.43	0.19 ± 0.13	NS	NS
**Proteobacteria**	15.91 ± 3.1	22.14 ± 6.47	NS	NS
*Escherichia-Shigella*	15.68 ± 3.07	21.98 ± 6.49	NS	NS
**Bacteroidetes**	18.64 ± 6.78	11.00 ± 2.21	NS	NS
*Bacteroides*	14.97 ± 6.01	9.85 ± 2.16	NS	NS
*norank_f_Muribaculaceae*	0.56 ± 0.45	0.53 ± 0.51	NS	NS
*unclassified_f_Prevotellaceae*	0.88 ± 0.87	0.02 ± 0.01	NS	NS
Actinobacteria	4.19 ± 1.66	8.01 ± 4.78	NS	NS
*Bifidobacterium*	2.67 ± 1.17	6.92 ± 4.83	NS	NS
*Eggerthella*	0.75 ± 0.47	0.72 ± 0.20	NS	NS

*n* = 7 for each group. L7PCO*N* = CON piglets on L7; L7PFFM = FFM piglets on L7; NS = No Statistic.

**Table 7 microorganisms-09-00604-t007:** The main microbial pathways are grouped into level 3 Kyoto Encyclopedia of Genes and Genomes (KEGG) functional categories.

Pathway Level 2	Pathway Level 3	G60 CON	G60 FFM	*p*
Carbohydrate metabolism	Glycolysis/Gluconeogenesis	7.87 ± 0.51 ^a^	7.37 ± 0.26 ^b^	0.039
	Citrate cycle (TCA cycle)	7.26 ± 0.2 ^a^	7.00 ± 0.22 ^b^	0.04
	Pentose phosphate pathway	3.74 ± 0.05	3.77 ± 0.03	0.121
	Pentose and glucuronate interconversions	3.27 ± 0.07 ^b^	3.35 ± 0.06 ^a^	0.025
	Fructose and mannose metabolism	2.85 ± 0.21	3.03 ± 0.11	0.07
	Galactose metabolism	2.86 ± 0.07 ^b^	2.93 ± 0.04 ^a^	0.058
	Ascorbate and aldarate metabolism	2.77 ± 0.17	2.91 ± 0.09	0.07
Lipid metabolism	Fatty acid biosynthesis	2.17 ± 0.10 ^b^	2.28 ± 0.07 ^a^	0.033
	Fatty acid elongation	1.96 ± 0.14 ^b^	2.09 ± 0.07 ^a^	0.045
	Fatty acid degradation	1.84 ± 0.09	1.83 ± 0.04	0.656
	Synthesis and degradation of ketone bodies	1.89 ± 0.1 ^a^	1.77± 0.08 ^b^	0.028
	Steroid biosynthesis	1.74 ± 0.03	1.74 ± 0.03	0.762
	Primary bile acid biosynthesis	1.67 ± 0.03	1.66 ± 0.05	0.426
	Secondary bile acid biosynthesis	1.59 ± 0.03 ^b^	1.63 ± 0.03 ^a^	0.021
Energy metabolism	Oxidative phosphorylation	1.43 ± 0.07	1.44 ± 0.02	0.777
	Photosynthesis	1.41 ± 0.03	1.43 ± 0.02	0.174
	Photosynthesis—antenna proteins	1.42 ± 0.02	1.4 ± 0.03	0.16
Amino acid metabolism	Alanine, aspartate and glutamate metabolism	1.3 ± 0.04	1.32 ± 0.03	0.431
	Cysteine and methionine metabolism	1.14 ± 0.03	1.13 ± 0.02	0.336
	Valine, leucine, and isoleucine degradation	1.11 ± 0.02	1.12 ± 0.03	0.383
	Valine, leucine, and isoleucine biosynthesis	1.1 ± 0.04	1.08 ± 0.03	0.408
	Lysine biosynthesis	1.05 ± 0.1	1.11 ± 0.06	0.196
	Lysine degradation	1.05 ± 0.08	1.02 ± 0.06	0.403

^a, b^ Means within rows with different superscript letters indicate significant differences (*p* < 0.05); *n* = 7 for each treatment. G60 CO*N* = the fecal microbiota communities of CON sows; G60 FFM = the fecal microbiota communities of FFM sows supplemented with 25.5% FFM.

## Data Availability

Not applicable.

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
