# Peer review of "Effects of the Inclusion of Fermented Mulberry Leaves and Branches in the Gestational Diet on the Performance and Gut Microbiota of Sows and Their Offspring"

_microorganisms, 2021, doi:10.3390/microorganisms9030604_

Round 1

Reviewer 1 Report

The author has already addressed majority of my previous concerns, so, I am ok to publish the manuscript in the present form. 

Author Response

reviwer 1

Point 1:The author has already addressed majority of my previous concerns, so, I am ok to publish the manuscript in the present form. 

Response: Thank you very much for your comments and suggetions.

Reviewer 2 Report

The title in its current form is adequate to the research carried out.
However, the keywords need to be changed as they are repeating the title. the most important references characterizing the subject of research. Keywords should be short and clearly formulated so that the reader can easily find the article in the search engine after mixing the most important terms.
The structure of individual chapters is correct. Some suggestions concern the abstract, which in my opinion should be slightly edited. Abstract: It's too general in content. Please leave the most important information without the descriptions that are included in the introduction and summary. English is very poor in some sections. It is necessary to verify the entire text by a native speaker. Moreover, the authors use colloquial language. Such wording is not appropriate for a research paper.
In table 1, the full wording should be used. The use of abbreviations causes considerable inconvenience due to the overly confusing and very extensive description under the table, which should be clear. Currently, it is difficult to find individual explanations for abbreviations because there are too many of them.
Lines 73-76, chapter: Materials and Methods. There is no information on how the nutritional indicators were determined. The authors should provide information on the basis of which the content of all ingredients in Table 1 was determined. While for the basic feed, the content of the ingredients can be assumed to have been supplied by the feed company, for the feed modified with the addition of fermented moiré (leaves and stalks) it is no longer clear. Therefore, the methods used to determine these components should be stated. Of course, with a detailed description of the methodological procedure. Bearing in mind that the origin of the chemical reagents should also be given.
The summary is too laconic and poor. Such an extensive experiment and extensive research results should lead to specific conclusions. Authors should formulate specific conclusions summarizing the experiment. The summary cannot be a repetition of the abstract and excerpts from the results chapter. Recommendations for practice should be provided. After all, that was the assumption.

Almost 1/3 of the references are items of the 10-year-old and older ponds.

Author Response

Reviewer 2

Point 1 The title in its current form is adequate to the research carried out. However, the keywords need to be changed as they are repeating the title. the most important references characterizing the subject of research. Keywords should be short and clearly formulated so that the reader can easily find the article in the search engine after mixing the most important terms.

Response: we changed the keywords “fermented feed mulberry” to “mulberry”, and add “gestation; lactation”. (line 26)

Point 2 The structure of individual chapters is correct. Some suggestions concern the abstract, which in my opinion should be slightly edited. Abstract: It's too general in content. Please leave the most important information without the descriptions that are included in the introduction and summary.

Response: We rewrite the abstract, in red (line14~25), but due to the 200-words limit, the new abstract may still not be very detailed.

Point 3 English is very poor in some sections. It is necessary to verify the entire text by a native speaker. Moreover, the authors use colloquial language. Such wording is not appropriate for a research paper.

Response: Before submitting the article, we have completed the language editor recommended by MDPI to modify the language (the proof is as follows), and we have also invited native experts to help modify the language. At the same time, due to the relatively long content of this article, there may still be some language flaws, we will continue to modify it

Point 4 In table 1, the full wording should be used. The use of abbreviations causes considerable inconvenience due to the overly confusing and very extensive description under the table, which should be clear. Currently, it is difficult to find individual explanations for abbreviations because there are too many of them.

Response: We changed the table 1, represents with the full wording (showed in red, line 105), and delete the description for abbreviations under the table(line 87~line 98).

Point 5 Lines 73-76, chapter: Materials and Methods. There is no information on how the nutritional indicators were determined. The authors should provide information on the basis of which the content of all ingredients in Table 1 was determined. While for the basic feed, the content of the ingredients can be assumed to have been supplied by the feed company, for the feed modified with the addition of fermented moiré (leaves and stalks) it is no longer clear. Therefore, the methods used to determine these components should be stated. Of course, with a detailed description of the methodological procedure. Bearing in mind that the origin of the chemical reagents should also be given.

Response: We divided the original 2.2 into 3 parts: the new 2.2, 2.3 and 2.4. And we described the determination method of the nutritional indicators of FFM and FFM supplementation feed in section 2.4, and the reagents were added also, see line 92~110 and line106.

Point 6 the summary is too laconic and poor. Such an extensive experiment and extensive research results should lead to specific conclusions. Authors should formulate specific conclusions summarizing the experiment. The summary cannot be a repetition of the abstract and excerpts from the results chapter. Recommendations for practice should be provided. After all, that was the assumption.

Response: We rewrite the conclusion,added the recommendation for practice.(line487~500)

Point 7 Almost 1/3 of the references are items of the 10-year-old and older ponds.

Response: We have changed and supplemented the new literature in 5 years, but because some references have classic reference value and cannot be replaced, there are still 11 documents that are more than 10 years old, and the total number of references is 72(line 521-714).

Round 2

Reviewer 2 Report

The authors made changes to the article according to the suggested recommendations.
They improved the methodology by adding the missing descriptions of the analytical procedure. All doubts have been resolved. The summary was modified to include the conclusions from the experiment. Literature was supplemented with new references. The authors have made every effort to ensure that the content is legible and clear for the recipient, therefore, based on the revision of the manuscript, I believe that it may be published in its current form.

This manuscript is a resubmission of an earlier submission. The following is a list of the peer review reports and author responses from that submission.

Round 1

Reviewer 1 Report

  1. There are several typos throughout the manuscript which need to be addressed (e.g. line 69 remove space after comma, 72 add space after comma etc.). 
  2. Line 84: Remove the sentence "During 21 days ....adlibitum) as you are writing the same thing in Line 92. 
  3. In materials and methods section: Please provide a table that summarizes different samples taken from sows and piglets at different time intervals to analyze different parameters. This will be very helpful for the readers to follow the different tests performed in different specimens at different intervals from sows and piglets; reading from the text is so hard to follow. 
  4. Fecal Microbiota analysis: 
    1. Did you use any controls? (for instance: negative control from extraction to the sequencing, positive control such as mock community). This is very important for quality control of each steps of microbiota analysis. 
    2. Details of 16S rRNA sequence analysis need to be provided. For instance: what did you use for quality trimming?, how you assign taxonomy?, which database you use for the taxonomic assignments? How you calculate alpha and beta diversity? All software/packages used including their versions must be provided. This is extremely important for robust and reproducible results. 
    3. Are you depositing the raw sequence data in public repository? I am not sure about the requirement by the journal, however, it is a requirement nowadays by almost every journals to publish the manuscript with sequencing data. 
    4. For statistical analysis of microbiome data: Did you use normalization of the data? For comparison of different taxa, what is the rationale of using statistical tests like Student t-test or Kruskal Wallis instead of using other tests that are especially designed for the microbiome data sets such as ANCOM, LEfSe etc. 

Reviewer 2 Report

  1. English languish is very poor in some sections
  2. The experimental design is not clear
  3. Lack of references
